# Developing Burdens in Caring for a Relative with a Cancer Diagnosis: A Qualitative Study of Lived Experiences of Family Caregivers in Saudi Arabia

**DOI:** 10.3390/nursrep15070233

**Published:** 2025-06-26

**Authors:** Eman Halil Al Enazy, Seham Mansour Alyousef

**Affiliations:** 1Psychiatric Department at Prince Sultan Military Medical City, Riyadh 14814, Saudi Arabia; 2Community and Psychiatric Department, Nursing College, King Saud University, Riyadh 11451, Saudi Arabia; smansour@ksu.edu.sa

**Keywords:** cancer, caregivers, family, caregiver burden, relative, qualitative study

## Abstract

**Background:** Cancer is a worldwide public health issue that impacts individuals in many ways. Family caregivers (FCGs) play a crucial role in providing care for cancer patients; as a result, they face several challenges as caregivers that sometimes go unreported. **Aim:** The purpose of this study was to explore the lived experiences of family caregivers with a developing burden of caring for a relative with a cancer diagnosis in Saudi Arabia. **Methods:** A qualitative study employing thematic analysis was conducted with ten family caregivers recruited through purposive sampling from the Prince Sultan Military City Hospital’s oncology department. Semi-structured interviews were conducted from July to August 2023 until data saturation was achieved. **Results:** Ten family caregivers participated in the study. Four main themes emerged from the data: antecedents to the caregiving burden, attributes of the caregiving burden, consequences of the burden, and religion and beliefs. Across these four main themes, there were 12 sub-themes. **Conclusions and Implications:** Family caregivers are crucial in caring for and supporting cancer patients. Thus, it is essential to shed light on family caregivers, who often remain invisible as secondary patients in healthcare systems, to understand the factors associated with developing caregiver burden. Education and support for family caregivers may decrease the burden on the family caregiver, which can positively impact the whole family unit, including the relative with the cancer diagnosis.

## 1. Introduction

Cancer is a collection of disorders characterized by uncontrolled cell growth and organ invasion. According to the World Health Organization (2024), by 2050, almost 35 million additional cancer cases are anticipated, representing a 77% rise over the projected 20 million cases in 2022 [1]. Worldwide, cancer kills 16% of people, making it the most significant public health issue [2]. In Saudi Arabia, cancer ranks third among the leading causes of death. The prevalence rate of cancer in KSA as a whole is 88.7 cases per 100,000 people [3].

Both cancer and cancer treatment have an impact not only on the patients themselves but also on their family members and caregivers. When providing care for individuals suffering from life-threatening diseases like cancer, caregivers face both physical and mental difficulties [4,5]. Several studies indicate that the repercussions of a cancer diagnosis have a more significant effect on family members compared to patients [6]. In the course of the disease, when cancer patients are not hospitalized, family caregivers have crucial responsibilities in providing patient support. However, the act of providing care does not come to an end during hospitalization [6].

Family caregivers are typically unpaid individuals, often family members, who assist with daily living activities and medical follow-ups for disabled children, as has been observed in Saudi Arabia, where family values and commitments are highly respected [7]. The caregiving burden refers to the physical, emotional, and financial strain experienced by caregivers, which can vary significantly based on cultural and socioeconomic factors [8,9]. This burden is compounded by a lack of formal support systems, as caregiving is often seen as an informal aspect of healthcare, leading to psychological distress and reduced quality of life [10].

The caregiving experience in Saudi Arabia is deeply influenced by cultural, familial, and religious factors, which are crucial in shaping the roles and expectations of caregivers. In the Gulf Cooperation Council (GCC) region, where caregiving is often rooted in Islamic cultural practices, family members, particularly women, are expected to provide care for relatives, reflecting traditional gender roles and familial obligations [11]. In Saudi Arabia, the burden on caregivers is exacerbated by limited formal social support systems, despite the significant impact on their quality of life. While families often share caregiving responsibilities among extended family members, social support remains constrained because of the following: (1) support is typically concentrated within the immediate family, without broader community resources; (2) female caregivers may experience additional burdens despite family help due to cultural expectations of primary caregiving responsibility; and (3) formal support systems remain inadequate, regardless of informal family assistance. Caregivers often experience high levels of stress and anxiety due to these inadequate formal support systems [12]. The role of religion, particularly Islam, is significant, as it not only influences caregiving practices but also serves as a coping mechanism for managing stress and emotional burdens [13]. The familial structure, often including the extended family, plays a critical role in caregiving, so that responsibility is shared among family members, yet primarily falls on women, reinforcing traditional gender roles [14,15,16].

Family caregivers of cancer patients experience significant physical, psychological, and social burdens [17,18,19]. These burdens include physical health problems, emotional stress, role conflicts, and financial strain [20,21]. Caregivers often face challenges balancing their caregiving responsibilities with personal and professional roles [22]. According to Williams et al. [23], characteristics of both patients and caregivers, along with the available social support, influence the caregiving experience. Recent healthcare shifts have modified traditional caregiving responsibilities. Factors such as caregiver self-esteem and gender—which have received less research attention—also significantly affect burden levels [24]. Further research is needed to explore the lived experience of family caregivers’ burdens in caring for a relative with a cancer diagnosis.

While existing studies have begun addressing various aspects of caregiver burden, significant gaps warrant further investigation. Firstly, while studies such as those by Saimaldaher & Wazqar [25] and Ghazwani et al. [26] have quantitatively assessed burden and stress among caregivers, qualitative insights are also needed which delve into the emotional and psychological dimensions of caregiving. For instance, the qualitative exploration of guilt experienced by caregivers carried out by Mohamed Hussin & Mohd Sabri [27] offers a glimpse into the emotional landscape of caregiving. However, this study did not explicitly address the distinctive cultural and social contexts of Saudi Arabia. This indicates a need for qualitative studies that focus on the nuanced emotional experiences of caregivers in this specific cultural setting, exploring themes such as the impact of cultural expectations on caregiving roles.

Experience is a subjective phenomenon through which individuals imbue meaning into their perception and their conscious response to a specific event or occurrence. Because experience is subjective, applying any metrics to evaluate an individual’s experience is impossible [28]. In Saudi Arabia, despite several studies on caregivers of cancer patients, attempts by researchers to assess the impact of formal care on these individuals have not effectively yielded empirical insights into the true character or substance of the caregiving experience. Quantitative studies cannot comprehensively depict caring experiences as effectively as qualitative research studies. Thematic analysis, a qualitative methodology, enables researchers to thoroughly investigate, explore, and articulate the real-life encounters of those who care for cancer patients while identifying patterns and themes within their experiences [29].

Moreover, caregivers’ experiences can be profoundly influenced by cultural beliefs and practices which may shape their coping mechanisms and support systems. For instance, the role of the Islamic faith in shaping the caregiving experience, as discussed in Hafez et al. [30], suggests that spiritual and religious dimensions are critical yet underrepresented. Future studies should explore how these cultural factors influence caregivers’ experiences and overall well-being.

Additionally, while some studies have documented challenges caregivers face, particularly in the context of palliative care, there needs to be more comprehensive qualitative research that captures the intersection of these challenges with the specific cultural and familial dynamics present in Saudi Arabia. There is a notable absence of qualitative research focusing exclusively on cancer caregivers in Saudi Arabia. This gap is critical, as the caregiving experience can differ significantly based on the type of illness, the stage of the disease, and the healthcare context. Qualitative research could provide valuable insights into how caregivers navigate these challenges and help to determine the support they require.

The burden of caring for a relative with a cancer diagnosis in Saudi Arabia is a significant issue that has not been thoroughly explored in qualitative research. While several studies have investigated caregiver burden in various contexts, there remains a notable gap in understanding the lived experiences of family caregivers, specifically for cancer patients in Saudi Arabia. This gap is critical, as caregivers often face unique challenges influenced by cultural, social, and economic factors. The existing literature indicates that the caregiver burden is multifaceted and encompasses emotional, physical, and financial strains. These challenges include high levels of stress and anxiety, limited access to formal support systems, and reliance on informal coping mechanisms, as has been highlighted in the context of the Gulf Cooperation Council region [11]. The emotional and psychological dimensions of caregiving, which are crucial yet underexplored, significantly affect caregivers’ quality of life. Caregivers often experience a negative correlation between their burden and quality of life, and this is exacerbated by inadequate social support [12]. The unmet supportive care needs of caregivers, particularly in developing countries like Saudi Arabia, include psychological distress and economic hardship which are further complicated by cultural and regional disparities [31]. Addressing these gaps through comprehensive research could lead to better support systems and interventions tailored to the unique cultural and socioeconomic context of Saudi Arabia. Therefore, this study explored the lived experience of the burden of family caregivers in Saudi Arabia whose relatives had cancer diagnoses. The findings identify culturally sensitive and holistic care pathways required to address the needs of Arab family caregivers. This research sought to give voice to this under-researched population.

## 2. Methods

### 2.1. Design

This qualitative study employed thematic analysis using a framework-informed approach to explore the lived experiences of family caregivers of patients with cancer diagnosis in Saudi Arabia. The analyzed data from semi-structured participant interviews, adhering to COREQ guidelines. This approach allows for an in-depth exploration of the lived experiences of family caregivers, capturing the essence of their caregiving journey [32,33].

### 2.2. Setting

The study was conducted in an adult oncology ward and a pediatric oncology day-case unit at Prince Sultan Military Medical City, a governmental hospital in the Kingdom of Saudi Arabia affiliated with the Ministry of Health and the Ministry of Defense and Affairs (MODA) in Riyadh City, KSA. This is a regional hospital with 192 beds which serves approximately 14,300 outpatients annually in all departments, with an average of 100 patients per month receiving cancer treatment and follow-up care.

### 2.3. Sampling

All participants (*n* = 10) were FCGs supporting loved ones through cancer treatment. To perform a thorough and relevant investigation of the issue of interest, researchers used the purposive sampling method to pick participants based on their characteristics [34,35]. This method is used to approach participants who are easily accessible and willing to participate [36]. In this study, participants who fitted the definition of FCGs and had been providing care for a relative with cancer were selected using a purposive sample method. Data saturation was systematically assessed following established guidelines. After the 8th interview, no new codes emerged from the data. Interviews 9 and 10 confirmed theoretical saturation, with themes well developed and no additional insights identified. This saturation point was documented through our analytical audit trail, which tracked code emergence across interviews. Data saturation is achieved when no new insights or information emerge from additional data collection [37]. The process involves identifying when further coding is no longer feasible, indicating that the data collected sufficiently covers the research questions and themes [37,38].

**Inclusion criteria:** being a relative or family member caring for a relative living with cancer; providing care for more than two weeks; being aged eighteen years or older; not having current mental health issues or a history of mental health problems; not being a healthcare professional in the same setting; being willing to take part in the study.

**Exclusion criteria:** being a formal caregiver; not being a relative or family member; being paid for caregiving; being a family caregiver of a patient in an unconscious condition; having a history of mental health problems not related to caregiving.

### 2.4. Data-Collection Instrument and Procedure

The data collection instrument employed in this study was a semi-structured individual interview guide. The interview guide comprised four open-ended questions designed to explore participants’ lived experiences as family caregivers (see Table 1). The questions aimed to explore antecedents to caregiver burden [39,40], attributes [39,41], and consequences of burden [41,42], and the role of religious and spiritual beliefs as coping mechanisms [40]. These questions were designed to align with the research aim of understanding the multifaceted nature of caregiver burden in the context of cancer care.

Family caregivers were approached while accompanying their relatives to the adult oncology ward or pediatric oncology day-case unit, serving as watchers and caregivers during hospitalization. After obtaining consent, semi-structured interviews were conducted over a period of four weeks, from July to August 2023. Each interview ranged from 60 to 120 min and was conducted in an open and exploratory manner. Table 1 presents the four main question areas. However, interviewers also asked extensive probing questions such as ‘Can you tell me more about that?’, ‘How did that make you feel?’, and ‘What was that experience like for you?’ to explore participants’ responses in depth.

### 2.5. Trustworthiness

To establish trustworthiness, we employed multiple strategies: (1) The interview guide was developed based on Liu et al.’s conceptual framework of caregiver burden, incorporating the four domains of antecedents, attributes, consequences, and coping mechanisms. A multidisciplinary team including mental health nursing faculty, a psychiatrist, a psychologist, a social worker, and a mental health nurse reviewed the guide for content validity and cultural appropriateness before implementation. (2) Questions were piloted with three caregivers to assess clarity and relevance, allowing for adjustments based on feedback. (3) Member checking was conducted with participants to validate emerging themes. (4) An audit trail documented all analytical decisions throughout the research process [43,44,45,46].

### 2.6. Ethical Considerations

The study obtained ethical approval from the IRB at KS University (E-23-8130) and approval from the Research Center at PSMMC, Health Ethics Committee Approval Reference (HP-01-R07). As a result, informed consent was obtained from participants for qualitative data collection. Participants were informed that anonymity would be maintained and that narrative excerpts would be anonymized using pseudonyms.

### 2.7. Data Analysis

The interview guide comprised four open-ended questions designed to prompt participants into recounting their lived experiences as family caregivers (see Table 1). Participants were invited to discuss their antecedents, the attributes of burden, the consequences of caregiving, and the religious and spiritual beliefs that influenced or aided their coping mechanisms. All interviews were audio-recorded and promptly transcribed on the same day or the day following. Observational notes were also taken during the interviews to capture verbal and nonverbal subtleties that emerged throughout the conversations. Both authors and a professional translator reviewed the transcribed interviews to ensure the accuracy of the translations from Arabic to English, maintaining the integrity of the expressed thoughts within the context of the interview.

Thematic analysis was carried out following the structured guidelines of Braun & Clarke [47], ensuring a rigorous and methodical approach. Both authors participated in the coding process. Initial codes were developed inductively from the data, with researchers independently coding the first three transcripts and then meeting to compare codes and develop a preliminary coding framework. Examples of initial codes included ‘financial strain’, ‘role conflict’, ‘physical exhaustion’, and ‘spiritual coping’. The process began with a thorough familiarization with the data; researchers immersed themselves by reading, re-reading, and listening to the transcribed interviews to grasp the depth of the content fully [47,48]. Initial coding was performed manually, marking significant text segments related to the study’s research questions. This initial coding was crucial for identifying potential themes. Following this, codes were sorted into potential themes through collaborative efforts where multiple researchers independently grouped codes and subsequently refined these categorizations through discussions. Themes were meticulously reviewed and defined in team meetings, ensuring they accurately represented the data.

To ensure the rigor and reliability of our coding process, several measures were implemented. Inter-coder reliability was ensured by having multiple researchers independently code the data, with any discrepancies being discussed until consensus was reached. Inter-rater consistency was achieved through the coding process, in which multiple researchers independently coded transcripts and compared their interpretations until consensus was reached, rather than through consistency in interview administration. We also engaged in member checking, where the developed themes were validated by the participants to ensure that these themes genuinely reflected their perspectives. An audit trail was maintained throughout the analysis, with all decisions and interpretations being documented, adding an extra layer of transparency and rigor.

Moreover, trustworthiness was established using the characteristics of credibility, dependability, confirmability, and transferability, as outlined by Guba et al. [49]. The accuracy of the transcriptions and audio recordings of the interviews, along with the thorough questioning during the interviews, contributed to establishing credibility. To confirm the subjective impressions from each interview, data collection continued until saturation was reached, and nonverbal cues and thoughts were recorded immediately after each interview. Iterative listening and thorough verification of information ensured dependability. Providing detailed explanations of the study procedure enhanced transferability, while confirmability was improved through discussions of results with the second author. These comprehensive steps collectively ensured that our thematic analysis was not only systematic but also robust, providing a trustworthy narrative of the participants’ experiences.

Throughout the study, ongoing comparison and analytical induction methods were employed, significantly enhancing the development of sub-themes and categories during the note-taking, transcription, and interpretation phases. Ongoing comparison involved continuously comparing new data with previously coded data to refine themes and identify patterns. Analytical induction involved systematically examining cases that did not fit emerging patterns to ensure comprehensive theme development. This approach ensured a thorough and nuanced understanding of the data. Identifying the researcher’s positionality is crucial in qualitative research, as it influences data collection and analysis. The researchers engaged in critical reflection on their positionality, acknowledging how social position shapes perception, following the insights of Temple & Young [50]. The lead researcher, a female psychiatric nurse with experience in oncology settings, conducted all interviews. We maintained reflexivity throughout the research process, acknowledging how researcher characteristics might influence data collection and interpretation. The researcher’s cultural background as a Saudi national facilitated understanding of cultural nuances and religious references made by participants, while we remained vigilant about potential bias through member checking and peer debriefing to ensure authentic representation of all participants’ voices [51].The researcher’s female gender affected participant engagement and communication, leading to an emotional investment in the study. Moreover, grounded on a solid theoretical and sociological framework, the researcher generated authentic representations of the participants’ viewpoints. Although researcher and participants shared a similar cultural background, a diversity of opinions was nevertheless recognized and well represented.

## 3. Results

### 3.1. Personal Data of Family Caregivers of Patients with a Cancer Diagnosis

Ten family caregivers participated in the study, aged 28–43 years. The majority were female (80%, *n* = 8), Saudi nationals (100%, *n* = 10), and married (90%, *n* = 9). Regarding education, 60% (*n* = 6) held high-school certificates. Socioeconomically, 40% (*n* = 4) were employed and had a monthly income above SAR 10,000. Additionally, 60% (*n* = 6) reported having domestic help for household tasks. All participants had government health insurance coverage. Regarding relationship to the cancer patient, 40% (*n* = 4) were daughters, 30% (*n* = 3) were mothers, and 10% (*n* = 1) each were father, son, or sister. The detailed sociodemographic data are shown in Appendix A.

Four main themes emerged from the data, which aligned with but also extended Liu et al.’s [52] conceptual framework: antecedents to the caregiving burden, attributes of the caregiving burden/consequences of the burden, and religion and beliefs. Notably, our analysis revealed a novel fourth theme—‘religion and beliefs’—which was not identified in Liu et al.’s concept analysis, representing a unique contribution specific to our Saudi Arabian context. Additionally, the analysis identified ‘unsatisfactory treatment’ as a new subtheme under attributes, reflecting cultural expectations for family involvement in medical decisions. These four main themes encompassed 12 sub-themes that provided detailed insights into the multifaceted nature of caregiver burden experiences (Figure 1 and Table 2).

### 3.2. Theme 1: Antecedents to the Caregiving Burden

The subthemes identified in this section pertained to insufficient financial resources, multiple responsibility conflicts, and lack of social activity.

#### 3.2.1. Insufficient Financial Resources

Findings revealed that most participants identified and elaborated on the financial burden due to caregiving to a relative with a cancer diagnosis. Insufficient financial resources may be due to lack of a job, or because the caregiver has to leave their job to be able to take care of their relative, or because of increased financial expenses due to caregiving demand. Compromised patients, especially during chemotherapy sessions, need special attention. The food they eat and the environment they live in should be checked for cleanliness and safety. Such needs may increase the expenses of family caregivers. Moreover, if the patient and the family member are originally from an area far from the treating hospital, the costs of transportation and of living beside the hospital in case of discharge may add further financial strain to the family. One participant said, “*I have to take unpaid leave to take care of her; I do not regret it if she asks for anything, we will do it from the heart, we just want her to gain her health back*” (P5). Another participant reported, “*It is hard at the beginning especially when my mother was newly diagnosed, and she has to travel to the hospital for seeking for the treatment we decided to go with her, some of us were able to travel by plane, and some travelled to by land, we do not care about financial spending’s as long as my mother will be okay*” (P7).

#### 3.2.2. Multiple Responsibility Conflicts

The family caregiver may be a wife or husband, mother or father, or daughter or son of someone else, before assuming the role of family caregiver. Three participants mentioned having children and leaving them with another family member so they could take care of a relative with a cancer diagnosis. The additional role of being a family caregiver alongside other roles and responsibilities may cause conflicts, leaving the caregiver stressed and feeling lost about what should and what should not be done. One participant said, *“We are lost. We do not know what we should do for my mother, we left everything behind us, my husband, my children, and I do not care for anything in my life except my mother now”* (P4).

Moreover, one participant decided to take care of her mother while the latter was hospitalized for around two months, leaving her own children with her single younger sister in their parents’ house. *“Priorities changed; my mother is the priority before my kids and myself”* (P9).

#### 3.2.3. Lack of Social Activity

Apart from the personal experiences of FCGs, it was noted that social support and activity and its relation to their burden was either positive or negative. Social support is a critical component of solid relationships and psychological health. It involves having family members and friends who can be turned to in times of need. Social support builds people up during stressful times and often makes them strong enough to carry on caring for a relative with a cancer diagnosis. Social support, such as phone calls, visitations, and offers of help, is only counted as social support by the family caregiver if they perceive it as support. Receiving support does not necessarily mean that the family caregiver perceives it. Social support, if perceived by family caregivers, might lessen the stress and burden of caregiving.

Long-term hospitalization affected one FCG negatively, as she was feeling alone because she and her husband used to live in Hafer al-Batin. The husband was still committed to his job. P2 said, *“My husband understands and feels what I’m going through, thus whenever he is able to come, he will come and will take us for out on pass for a day in a hotel to change my mood by changing environment”.*

Other comments made by participants in this regard included the following:

“With regards to social support, it is important, but it did not help me at the beginning with the stress of the diagnosis and moving from the city, leaving my home and other kids, and taking a loan to rent a suitable flat near the hospital made me more dispirit, having crying bouts and being depressed.” (P6)

“We have a perfect social support system, my family all around here they came to Riyadh to support my mother and us” (P7).

“Me and my two brothers are with him constantly, we are not leaving him, for more than 4 months we are leaving everything and staying with our father.” (P10)

### 3.3. Theme 2: Attributes of the Caregiving Burden

During the interviews, certain aspects of caregiving emerged repeatedly. These may be understood as critical attributes of the caregiver burden; in other words, characteristics or events related to the caregiving role.

#### 3.3.1. Self-Perception

Participants expressed their self-perceptions or feelings about themselves caring for a relative with a cancer diagnosis. Self-perception was understood as how caregivers perceived themselves while caring for a relative with a cancer diagnosis or the feelings they had about themselves and their new role as caregivers. Positive self-perception (feeling capable and satisfied with caregiving) was associated with a lower perceived burden, while negative self-perception (feeling inadequate or overwhelmed) corresponded with higher burden levels, as evidenced by contrasting participant statements. One participant expressing positive self-perception stated, ‘My feeling about my experience is alhamdulillah good’ (P5), while another participant expressing negative self-perception said, ‘We are only watching my mother dying’ (P4). These contrasting perspectives emerged from participants’ narratives about their caregiving experiences, with those expressing competence and spiritual acceptance reporting less distress compared to those expressing helplessness and despair.

#### 3.3.2. Multifaceted Strain

Due to the role of caregiving having various aspects, including hospitalization and long-term caring, the experiences of family caregivers may have multidimensional strains. As reported by participants, these may include and not be specific to problems in relationships due to lack of socialization, neglect of health, sleep disturbance, and fatigue.

“I had to leave my other child, the twin of my sick child, to stay for a long time hospitalized and caring for my sick son, and one time I had a sensation that my other child was sick. I had chest tightness as well when I called my mother, and she checked my child; my child was sick and gasping for breath, and my parents rushed him to hospital.” (P2)

Moreover, caring over a long period involving assumption of the role of family caregiver has the consequence of impacting other familial roles. The caregiver is a mother or father, a daughter or son, or a wife or husband before assuming the role of family caregiver. Being a family caregiver to a relative with a cancer diagnosis, especially during the hospitalization period, can lead to multiple responsibility conflicts, in addition to adding more strain and burden to the family caregiver, affecting them negatively.

“I have another problem. My daughter has been admitted now to King Fahad Medical City, and my husband told me that they suspect leukemia; since then, I am more stressed and worried, but what can I do except to Allah.” (P7)

#### 3.3.3. Long-Term Care

Long-term care has either a negative or positive impact on the caregiver’s burden. Many family caregivers reported that caring affected them negatively, as they felt alone and isolated. However, some participants stated that, over the long term, they were able to adjust so that their coping improved.

“Alhamdulillah’s now is too much better in comparison to the first month” (P8).

#### 3.3.4. Unsatisfactory Treatment (New Sub-Theme)

Interviews with four particular participants resulted in our identifying a new sub-theme among the attributes of caregiver burden; this was titled unsatisfactory treatment, and referred either to the treatment plan, as mentioned by some, or to the treatment team, including the primary physician responsible for the treatment plan.

*“The doctor refused to start up any treatment for my mother, as he said she would not be able to tolerate it. We are just sitting here watching her die, and we cannot do anything”* (P3).

“The team is not consulting us with the treatment; as you can see, our mother is sedated with strong meds; she does not need it because she was not in pain, but still, the treating team is prescribing it to her, and with no other interventions, we cannot speak to her, and she is not aware of us.” (P4) and (P7)

“My father is constantly vomiting, and we do not know how to help him; even the doctor decided to discharge him as he is a palliative case, but how can I discharge him while no medication is stopping nausea and vomiting? My father is not comfortable, but we cannot do anything.” (P10)

### 3.4. Theme 3: Consequences of the Caregiving Burden

Consequences are the results of caregiving to a relative with cancer, either for the patient or for the caregiver. These include the negative outcomes of caregiving. Such consequences include decreased care provision, decreased quality of life, and physical health deterioration.

#### 3.4.1. Decreased Quality of Life

As discussed by three participants during semi-structured interviews, long-term caring for a relative with a cancer diagnosis, along with other constraints, may decrease the quality of life of the family caregiver. In addition, decreased socializing with others resulting from hospitalization of a relative may also contribute to decreased QOL. Constant caregiving may lead to deterioration in physical health. In addition, hospitalization decreases the level of social activity enjoyed by the caregiver. Moreover, a family caregiver during hospitalization is less likely to participate in family gatherings. Participants reported that decreased social interaction and other elements affected the quality of life of both patients and caregivers.

“It was harder for my wife as she was the one admitted with our son; she got physically tired and not sleeping well.” (P1)

One participant lived in Riyadh with her husband, but their shared family lived in the south of the kingdom of Saudi Arabia; as a result, her social activity had decreased.

She said, “I am living with my husband in Riyadh. Our family is in the south of Saudi, but I have my cousin, and they used to visit me also but not frequently” (P8)

Furthermore, long-term caring and looking after a relative with a cancer diagnosis affects the quality of life of the family caregiver in terms of both psychological and physical health. Family caregivers, because of their caregiving role, may assist their patient relative in the activities of daily living. Attending to the needs of a cancer patient may also affect the sleeping pattern of the caregiver, either by insomnia related to worries or decreased sleep quality because of frequent interruptions. Giving more priority to the needs of the patient than those of the caregiver may affect the health of the latter by causing them to forget their own medical needs, as (P5) said:

“I am not sleeping well, and sometimes I forget even to take my antihypertensive meds because my sister is the priority; I give all my time to caring for her.”

#### 3.4.2. Decreased Care Provision

During interview, one of the ten participants highlighted a decrease in care provision. Decreased care provision for the patient happens when the family caregiver is burdened with the caregiving role. The family caregiver may be physically tired and unable to fully assess the patient as a result of their own daily life activities; they may forget an appointment or fail to administer medications to a relative with a cancer diagnosis as a result of decreased concentration. Alternatively, a caregiver may fail to focus on patient response if fatigued.

“One time I forgot to give her the medication, and I gave it late, and I usually forget to take my medication.” (P5)

#### 3.4.3. Physical Health Deterioration

During individual interviews, two participants mentioned physical health deterioration in response to caregiving to a relative with a cancer diagnosis. Assuming the role of caregiver to a relative with a cancer diagnosis can be physically exhausting. The family caregiver usually assists the relative with cancer in performing activities of daily living like eating, showering, dressing, attending follow-up appointments, and taking medication. The family caregivers may have no time for themselves to take rest periods or to eat well. If they themselves have physical health problems, they may even fail to take their own medications and attend their own follow-up appointments due to the demands and responsibilities of caregiving to a relative with cancer. Such physical health deterioration may increase during hospitalization, especially if no other family member is supporting and helping.

Physical health deterioration is reported as being tired of caregiving, feeling fatigued, and experiencing poor sleep quality due to hospitalization and its related frequent procedures and interventions. *“It was harder for my wife as she was the one admitted with our son; she got physically tired and not sleeping well.”(P1)*

Moreover, long-term hospitalization and caring for a patient with a cancer diagnosis who in some situations needs full assistance can be difficult. In addition to psychological health deterioration, physical health deterioration may occur as a result of prolonged caregiving. Some participants reported changes in sleeping patterns; some lost weight due to decreased dietary intake. “Even if I am not sleeping well, who will not be tired if he is watching a sick person admitted to the hospital!” (P9).

#### 3.4.4. Psychological Health Deterioration

Some FCGs also reported psychological health deterioration beginning at the time when the family caregiver first heard the diagnosis of cancer. A cancer diagnosis to a family member or relative can be associated with anxiety, a sad mood, or a depressive mood. In addition, long-term caregiving by a family care provider, such as caring for a relative with a cancer diagnosis, can be stressful. Long-term hospitalization, feeling isolated, and the inability to perform social activities may also contribute to psychological health deterioration.

“It was hard at the beginning; we were all psychologically stressed, especially when chemotherapy sessions started” (P1).

“It was harder at the beginnings, but know alhamdolelah”. Moreover, “I used to have depression, and crying all the time, seeing the life is black and not worthy, and my daughter will die after all of this” (P8).

Participant (P2) described her experience as “painful “due to multiple factors including long-term time hospitalization and caring for a family member with a cancer diagnosis affecting psychological health. “It is a harrowing experience”, she stated, adding the following:

“I am pretending to be strong, especially in front of my parents. I do not want to stress them more, but the moment I close the phone, I will cry deeply and show my weakness only to Allah; he is the only one who can resolve the situation.” (P2).

In addition, fast deterioration in cancer patient health can be traumatic, affecting the family members’ quality of life, and may affect physical and psychological health, causing deterioration in either or both.

“As you can see, everything changed, and my mother’s health deteriorated very fast since the diagnosis. Around one month, our life changed” (P3).

“Made me more dispirit, having crying bouts and depressed.” (P6).

Caring for a loved one with a terminal disease can be associated with overwhelmed emotions; the caregiver might be dispirited, hopeless, anxious, having a depressed mood, and sometimes crying, as reported by participants.

“Me and my all siblings, if we go out of the room, we will cry together and talk about our situations and struggles, but when we have to go back to my mother’s room, we have to wear another face to make her comfortable, because if we are stressed, my mother will get stressed” (P7).

“It is not easy to hear the one you love is affected by terminal cancer, but we are, as Muslims, accepting whatever from Allah” (P9).

### 3.5. Theme 4: Religion and Beliefs (New Theme)

When a person is stressed, this may be expected to affect them negatively; in rare cases, however, they will show a positive attitude, deeply accept the stress, and act positively. In Islam, accepting Allah’s plan is part of being faithful to Allah.

#### Religious Faith and Resilience

Being Muslim and having faith in Allah can bring some reassurance and help in coping with stressors and accepting and dealing with challenges in a realistic and calm way. “Alhamdolelah, everything from Allah we thank him for it, we are patient, and we are accepting it and no challenges” (P5).

One participant reported that her experience improved, and that she was able to cope, after she became more religious and closer to Allah. “There is a big hope, in such situation the human become powerless, and the only way to become more vital to be faithful to Allah and to become close to Allah, Allah is generous and robust” (P8).

The findings of this study confirm that family caregivers of a relative with a cancer diagnosis are subject to a developing burden in response to the caregiver role and to other factors, but that the burden lessens over time. In addition, the treatment plan of a relative with a cancer diagnosis may act as an attribute of the caregiver burden. Moreover, the participants in this study reported a new theme related to religion and belief that may have helped them cope with the multidimensional stress which they faced.

## 4. Discussion

This research aimed to investigate the experiences of family caregivers looking after a relative with cancer in Saudi Arabia. Additionally, the study aimed to identify the factors contributing to the burden experienced by family caregivers, such as financial strain, conflicting responsibilities, and limited social engagement.

Consistent with previous research [52], we found that family caregivers experienced financial strain, conflicts between various responsibilities, and a lack of social engagement as significant causes of caregiving burden. As cancer is a life-threatening illness, the length and negative consequences of treatment, as well as the patient’s inability to engage in work and social activities, result in significant changes in family dynamics and increased responsibilities and burdens for caregivers [53]. The “financial strain” theme may be attributed to employed family caregivers feeling compelled to quit their jobs, retire, or make unplanned changes to their work environments [54]. Similarly, and as also described in this study, numerous studies have shown that a caregiver’s financial resources can impact their role [55]. Recent research has indicated that a lack of essential financial resources [56,57], as well as inadequate income to meet caregiving demands [56,58], also contributes to the financial burden faced by caregivers. Additionally, 21 percent of caregivers in the United States report experiencing financial difficulties due to their role as family caregivers [59].

Our findings reveal significant variations in caregiving experiences based on socioeconomic status. Participants with higher incomes (*n* = 4, earning > SAR 10,000 monthly) reported an ability to hire domestic help which reduced physical caregiving demands but did not eliminate emotional burden. As P7 noted, “We have perfect social support system, my family all around here they came to Riyadh to support my mother and us,” reflecting how financial resources enabled family mobilization. Conversely, participants with lower incomes faced compound stresses of financial strain alongside caregiving responsibilities, as evidenced by P6, “taking a loan to rent a suitable flat near the hospital made me more dispirit, having crying bouts and being depressed”. Despite these differences, all participants regardless of socioeconomic status experienced similar emotional and psychological challenges, suggesting that certain aspects of caregiver burden transcend economic circumstances while others are clearly exacerbated by financial constraints.

The study findings indicate that participants recognized “multiple responsibility conflicts” as significant. Family caregivers provide care and handle a wide range of other tasks. They may be spouses, parents, children, or other relatives, and taking on the role of caregiver alongside other responsibilities can be challenging. Consistent with the findings of Stamataki et al. [60], this study reports that caregiving involves various responsibilities, including direct care, assistance with daily activities, emotional support, and medication monitoring. Many articles have shown that most family caregivers are multitasking spouses, children, or relatives.

According to previous research by Goldstein & Johnson [61], family caregivers are more likely to feel burdened due to their extensive caregiving, leading to restricted social networks. Long-term hospitalization hurts them, as they may lack family and friends to turn to in times of need, and relatives are more likely to offer support out of obligation rather than genuine curiosity about the situation. Studies by Noble et al. [62] and Sajjadi et al. [63] indicate that family caregivers experience burdens in their relationships as patients near the end of life, especially as care dependency increases.

The results of the study suggest that family caregivers experience both positive and negative emotions while caring for a cancer-diagnosed relative or taking on a new caregiving role. Similarly, as Girgis et al. [6] and Hodge & Sun [64] noted, caregiving experiences can be positive or negative. Studies have recommended examining factors such as the nature of the pre-disease relationship between the family caregivers and the patient and the availability of resources for providing care [65,66]. In a mixed-methods study on caregiver burden, De Korte-Verhoef et al. [67] found that more than half of family caregivers experienced a high burden.

According to the participants’ reports, family caregivers may experience a variety of strains as a result of their caregiving responsibilities and sporadic hospitalization, including relationship issues due to a lack of socialization, health neglect, sleep disturbances, and fatigue. Yoon et al. [68] described similar challenges. Furthermore, the findings of the study align with those of Ateş et al. [69], Arian et al. [70], and Lee et al. [71], indicating that caregivers of patients with cancer often suffer from health issues such as weight loss, fatigue, and sleep disturbances.

The results of the study also reveal that long-term caregiving can have either a negative or positive impact on caregiver burden. Hu et al. [72] suggested that the duration of caregiving, the level of social/family support, and the disease trajectory all significantly affect the burden on family caregivers. A longitudinal study in Taiwan indicated that overall burden levels perceived by family caregivers change over time, and that having another family member in need of care, or not having anyone to share the care task, was significantly correlated with changes in caregiver burden [71]. Conversely, factors such as deterioration in the functional status of patients and longer durations of caregiving are significant predictors of caregiving burden [73].

The findings revealed that one of the attributes of caregiver burden is unsatisfactory treatment, such as an unsatisfactory treatment plan or a treatment team that lacks a responsible primary treating physician. However, Naoki et al. [74] reported that the primary causes of dissatisfaction among family members included the information provided about prognosis, family conferences with medical professionals, and the involvement of family members in care decisions. There is a suggestion that patients diagnosed with cancer have a high level of unmet needs, particularly in terms of psychological support and medical information [75].

The findings of the study reveal that family caregivers often experience a decreased quality of life when caring for a relative with a cancer diagnosis, this decrease being associated with long-term caregiving responsibilities, decreased socialization due to hospitalization, and other constraints. This is consistent with a recent study by Cengiz et al. [76]. The effect of caregiver burden on quality of life varies depending on the phase of illness the recipient is experiencing, as indicated by Ochoa et al. [77]. Additionally, factors such as the caregiver’s age, their relationship with the patient, and socioeconomic status have been identified as influencing the level of burden experienced, suggesting that these factors may also interact with caregiving duration to affect outcomes [78].

The study’s findings also suggest that a reduction in care provision is one of the consequences of caregiver burden. Family caregivers reported feeling physically tired, which led to difficulties in providing adequate care, such as assisting with daily activities and managing medications. This is consistent with the findings of Adelman et al. and Choi & Seo [79,80]. The family caregivers reported experiencing physical health deterioration, including fatigue and poor sleep quality due to the demands of caregiving. Additionally, they also experienced psychological health deterioration from the moment they heard the cancer diagnosis of their relative. This is consistent with Gallego-Alberto et al.’s and Song et al.’s [81,82] findings.

The participants reported stress associated with caregiving as a first significant sub-theme. Participants also revealed that religious and spiritual beliefs interact dynamically with other aspects of caregiving. These beliefs not only influence caregivers’ resilience but also provide a source of comfort and a framework for understanding their caregiving role, making it essential to treat them as a separate theme. Additionally, the “religion and beliefs” theme emerged from the discussions, highlighting how religious beliefs and practices can serve as coping mechanisms for individuals facing stress and caregiving responsibilities. This aligns with the findings reported by King et al. [83].

Our findings align with international studies while at the same time revealing unique cultural dimensions. Unlike in studies conducted in the West, where formal support services are more readily available [61], our Saudi participants relied heavily on extended family networks and religious coping mechanisms. This contrasts with hospital-based caregiver support programs documented in European contexts [71], and highlights the need for culturally adapted interventions in Saudi healthcare settings. Our identification of ‘unsatisfactory treatment’ as a caregiver burden attribute extends beyond previous reports in the literature, suggesting that in contexts where family involvement in medical decisions is culturally expected, communication gaps between healthcare teams and families create an additional burden. Based on these findings, we recommend establishing multidisciplinary support teams including psychiatric nurses, social workers, spiritual care providers, and financial counselors specifically trained in cultural sensitivity for Saudi family caregivers.

Healthcare providers should adopt a culturally sensitive and holistic approach to supporting family caregivers, focusing on key strategies such as providing educational and emotional support to help manage anxiety and depression, especially during the later stages of caregiving. Additionally, targeted interventions should be implemented early on to address financial strain, role conflicts, and social isolation. It is also essential to recognize and incorporate cultural and religious coping mechanisms into caregiver support programs, ensuring that these interventions are tailored to the unique needs of caregivers in their specific cultural context. Furthermore, it is crucial to highlight potential interventions based on caregiving duration, such as offering support and resources early in the caregiving process to prevent burnout, or targeting interventions at critical stages in long-term caregiving to maintain caregivers’ health and well-being.

## 5. Limitations

The wide range of caregiving durations (2 weeks to several months) among participants may have influenced their perspectives and experiences. We chose the 2-week minimum to ensure participants had sufficient exposure to caregiving demands, while recognizing that this criterion might not capture the full spectrum of long-term caregiving experiences. The varying durations of caregiving experience represent a limitation, as caregivers in early stages may report different challenges compared to those with extended caregiving experience. In addition, our use of purposive sampling may limit the findings of this study so that they are not generalizable to all family caregivers in Saudi Arabia. Consequently, the results may not apply to all cancer patients in Saudi Arabia.

Furthermore, the research focused primarily on FCGs and their experiences, without including the opinions of cancer patients. Incorporating cancer patients’ experiences would have produced a more complete knowledge of the entire effect of cancer on people and their families in Saudi Arabia. Furthermore, the research only looked at FCGs’ experiences within a specific time range, limiting its potential to capture long-term impacts or changes over time.

Despite these limitations, the study provides valuable insights into caregiving experiences within the sampled population. Future studies should attempt to include a more extensive variety of participants and data-gathering methodologies in order to acquire a more representative and thorough picture of the experiences of people living with cancer in Saudi Arabia. Conducting qualitative research involving in-depth interviews with caregivers who have pre-existing mental health issues could uncover nuanced perspectives and complex emotional dynamics. This would allow us to examine if and how pre-existing mental health conditions influence the caregiving experience.

### Implications for Nurses

The findings presented here have several implications for nursing practice, emphasizing the critical need for comprehensive support systems for family caregivers. These caregivers often face considerable physical, psychological, and socioeconomic challenges that can lead to caregiver burden. Nurses play a crucial role in alleviating these burdens through psychoeducation, supportive care, and cognitive behavioral strategies. The complexities of caregiving, especially in palliative care settings, require nurses to assess family dynamics and address issues such as inadequate support and relational conflicts that can heighten caregiver stress. Our findings also suggest a need for an interdisciplinary, family-centered approach that highlights the importance of evaluating psychosocial needs and facilitating appropriate services, including palliative care consultations and support groups. By recognizing caregivers as integral members of the healthcare team, nurses can help improve their quality of life and resilience, ultimately enhancing both the caregiving experience and patient outcomes. In conclusion, this research underscores the importance of a holistic approach in nursing practice to support family caregivers, addressing their diverse needs and promoting positive results for both caregivers and patients. Finally, this research emphasizes the need for specialized support programs that meet the particular issues that these FCGs confront.

## 6. Conclusions

Family caregivers play a crucial role in caring for and supporting cancer patients, yet they often remain “hidden patients” whose needs require attention. Understanding the factors associated with caregiver burden is essential for supporting these vital care providers. This study reveals that caregivers experience significant multifaceted burdens. Physically, caregivers face health problems and exhaustion from caregiving demands. Psychologically, caregivers face substantial levels of emotional stress as well as mental health challenges. Socially, they encounter role conflicts, financial strain, and difficulties balancing caregiving responsibilities with their other life roles. Additionally, religion and beliefs significantly influence the caregiving experience, adding another layer of complexity to the situation faced by caregivers.

Our findings confirm that family caregivers (FCGs) are integral to cancer treatment and care, making their education and support paramount. When caregiver burden is reduced through appropriate interventions, the positive effects extend beyond the individual caregiver to benefit the entire family unit, including the cancer patient. This study contributes to the growing body of literature on caregiver burden by identifying culturally specific factors within the Saudi context and proposing a framework for culturally sensitive interventions. This cultural specificity is crucial for developing effective support systems that acknowledge the unique religious, social, and familial dynamics that influence caregiving experiences in this context.

## Figures and Tables

**Figure 1 nursrep-15-00233-f001:**
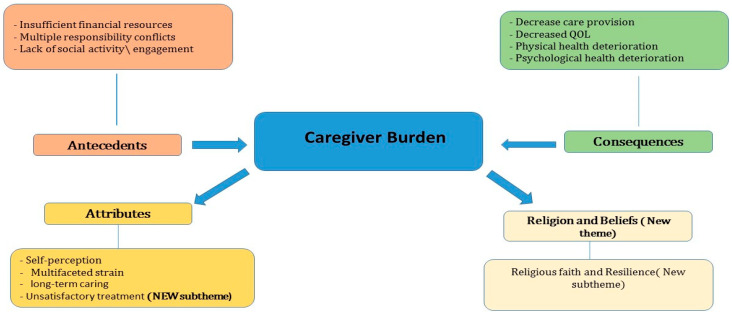
The new conceptual model of the caregiving burden.

**Table 1 nursrep-15-00233-t001:** Interview questions for participants.

1. How do the burden antecedents among family caregivers facilitate family caregiver burden development in caring for relative individuals with a cancer diagnosis?
2. Describe how the attributes of burden are associated with caregiver burden among family caregivers caring for a relative with a cancer diagnosis.
3. To what extent are the consequences of caregiver burden experienced by family caregivers caring for a relative with a cancer diagnosis?
4. To what extent can religious and spiritual beliefs affect the individual or serve as coping mechanisms for individuals facing stress and caregiving responsibilities?

**Table 2 nursrep-15-00233-t002:** Themes and sub-themes generated.

Themes	Sub-themes
Antecedents to caregiving burden	Insufficient financial resources Multiple responsibility conflicts Lack of social activity
Attributes of caregiving burden	Self-perception Multifaceted strain Long-term caring Unsatisfactory treatment
Consequences of caregiving burden	Decreased quality of life Decreased care provision Physical health deterioration Psychological health deterioration
Religion and beliefs (new theme).	Religious faith and resilience (new subtheme)

## Data Availability

The datasets utilized and/or analyzed in this study can be obtained from the corresponding author upon reasonable request.

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
