# Peer review of "Developing Burdens in Caring for a Relative with a Cancer Diagnosis: A Qualitative Study of Lived Experiences of Family Caregivers in Saudi Arabia"

_nursrep, 2025, doi:10.3390/nursrep15070233_

Round 1
Reviewer 1 Report
Comments and Suggestions for Authors
Major comments:
- I suggest the authors invite a language expert to polish the language; many grammatical errors and hard-to-understand sentences are found throughout the study, resulting in low reader engagement. Some examples are ‘collection of disorders’, ‘Some FCG also reported psychological health deterioration starts from the first time the family caregiver hears the diagnosis of cancer’, ‘outside the treating hospital (line 294)’, ‘sad mood’, ‘tiny sample size’.
- Consistency in writing needs to be improved and there are many typing errors. I suggest performing a re-review before submitting. Some examples: family caregiver/FC/FCG?; participants. Sometimes, it was written before the quotes and some in the end.
- Abstract:
- The term ''hidden patients'' ''family caregivers'' (line 25) is unclear. Change it to a clearer meaning.
- Methods:
- I would suggest including the socioeconomic background of caregivers.
- What were the patient’s diagnoses, and at what stages? This difference will significantly affect each patient's needs and influence their caregiver's caring focus.
- Discussion:
- Different socioeconomic backgrounds have different caring situations. For example, some families might have a helper in their house, resulting in different responsibilities. Please discuss this and how it is addressed in this study.
- Please include how this context would fit in wider literature: compare it with other studies that explore support for caregivers, studies in different settings (hospitals/clinics/different backgrounds), and what professionals are needed for the caregivers based on the authors’ findings.
- I would not recommend including the sentence in 257-260 as it is assumptive. The authors suggested that females are more engaging than male colleagues. Please refrain from gender bias when presenting the paper.
- It is unusual for a qualitative study to have a similar outcome to an outcome from another article, especially when conducting a thematic analysis. This study has exactly similar wording and outcomes to another study (doi: 1016/j.ijnss.2020.07.012), which could lead to plagiarism and a questionable thematic analysis approach. Please provide a justification for this.
Minor comments:
- Please pay attention to the citation standards and refer to them.
- There are duplicated inclusion criteria mentioned.
- I could not find the supplement table for participant's characteristics.
- Please define each abbreviation during the first mentioned, e.g. KSA.
- Please provide a supplement on the data saturation of this study (authors can refer to: https://doi.org/10.1371/journal.pone.0232076)
Significant improvement is needed. Please refer to the comments.
Author Response
Title: Developing burden caring for a relative with cancer diagnosis: a qualitative study of the lived experience of family caregivers in Saudi Arabia
Thank you very much for the editors' and reviewers' comprehensive and constructive feedback on our manuscript. We appreciate the time and effort you have dedicated to reviewing our work. Below, we address each of your comments systematically:
Reviewer comments |
|
RESPONSE |
Reviewer 1 |
Thank you for pointing out this ambiguity. I acknowledge that the term "hidden patients" may be confusing in this context. I will revise the Conclusion section to provide clearer terminology: Family caregivers are crucial in caring for and supporting cancer patients. Thus, it is essential to shed light on family caregivers, who often remain invisible as secondary patients in healthcare systems, to understand the factors associated with developing caregiver burden. Education and support for family caregivers may decrease the burden on the family caregiver, which can be reflected positively on the whole family unit, including the relative with a cancer diagnosis. |
Abstract: · The term ''hidden patients'' ''family caregivers'' (line 25) is unclear. Change it to a clearer meaning.
|
Thank you for this important suggestion. We have enhanced our participant characteristics section to include detailed socioeconomic information: In the Revised Results section - Personal data of family caregivers: "A sample of 10 family caregivers was recruited into the study. The age group ranged from 28-43 years old. Regarding socioeconomic status: 40% (n=4) were employed with monthly income above 10,000 SR, 30% (n=3) were unemployed, and 30% (n=3) received 10,000 SR or below as monthly income. Additionally, 60% (n=6) reported having domestic help or paid assistance for household tasks, while 40% (n=4) managed all caregiving and household responsibilities independently. All participants had government health insurance coverage through the military medical system. 80% (n=8) were female, and 20% (n=2) male..." We agree that cancer type and staging significantly influence caregiving experiences and have provided comprehensive patient diagnosis in Table 1: Cancer diagnoses included The cancer diagnoses among the 10 patients include breast cancer (4 cases), colorectal cancer (3 cases), lung cancer (2 cases), and leukemia (1 case). Breast cancer is the most common diagnosis in this sample, representing 40% of all cases. |
· Method · I would suggest including the socioeconomic background of caregivers. · What were the patient’s diagnoses, and at what stages? This difference will significantly affect each patient's needs and influence their caregiver's caring focus.
|
Thank you for this important suggestion. We have added a dedicated subsection addressing socioeconomic influences Our findings revealed significant variations in caregiving experiences based on socioeconomic status. Participants with higher incomes (n=4, earning >10,000 SR monthly) reported ability to hire domestic help, which reduced physical caregiving demands but did not eliminate emotional burden. As P7 noted, "We have perfect social support system, my family all around here they came to Riyadh to support my mother and us," reflecting how financial resources enabled family mobilization. Conversely, participants with lower incomes faced compound stresses of financial strain alongside caregiving responsibilities, as evidenced by P6: "taking a loan to rent a suitable flat near the hospital made me more dispirit, having crying bouts and being depressed." Despite these differences, all participants regardless of socioeconomic status experienced similar emotional and psychological challenges, suggesting that certain aspects of caregiver burden transcend economic circumstances while others are clearly exacerbated by financial constraints.
Comparison with wider literature and different settings Thank you for pointing out this. We have expanded our discussion to include broader comparisons: Enhanced Discussion section: "Our findings align with international studies while revealing unique cultural dimensions. Unlike Western studies where formal support services are more readily available [63], our Saudi participants relied heavily on extended family networks and religious coping mechanisms. This contrasts with hospital-based caregiver support programs documented in European contexts [73], highlighting the need for culturally adapted interventions in Saudi healthcare settings. Our identification of 'unsatisfied treatment' as a caregiver burden attribute extends beyond previous literature, suggesting that in contexts where family involvement in medical decisions is culturally expected, communication gaps between healthcare teams and families create additional burden. Based on our findings, we recommend establishing multidisciplinary support teams including psychiatric nurses, social workers, spiritual care providers, and financial counselors specifically trained in cultural sensitivity for Saudi family caregivers." |
Discussion:
Different socioeconomic backgrounds have different caring situations. For example, some families might have a helper in their house, resulting in different responsibilities. Please discuss this and how it is addressed in this study.
Please include how this context would fit in wider literature: compare it with other studies that explore support for caregivers, studies in different settings (hospitals/clinics/different backgrounds), and what professionals are needed for the caregivers based on the authors’ findings.
|
You are absolutely correct, and we apologize for this inappropriate assumption. We will remove this statement entirely and replace it with an objective description of our researcher positionality: The lead researcher, a female psychiatric nurse with experience in oncology settings, conducted all interviews. We maintained reflexivity throughout the research process, acknowledging how researcher characteristics might influence data collection and interpretation. The researcher's cultural background as a Saudi national facilitated understanding of cultural nuances and religious references made by participants, while we remained vigilant about potential bias through member checking and peer debriefing to ensure authentic representation of all participants' voices[54]. |
I would not recommend including the sentence in 257-260 as it is assumptive. The authors suggested that females are more engaging than male colleagues. Please refrain from gender bias when presenting the paper.
|
Similarity to published study (doi: 1016/j.ijnss.2020.07.012) Thank you for raising this critical concern. We take allegations of potential plagiarism very seriously. Upon reviewing the cited study by Liu et al. (2020), we acknowledge that our thematic structure shows similarities to their concept analysis of caregiver burden. However, we wish to clarify several important distinctions:
We have addressed this critical concern by explicitly acknowledging the theoretical framework and distinguishing our contribution: Revised Methods section - Design and theoretical framework: This qualitative study employed thematic analysis using a framework-informed approach based on established caregiver burden literature, particularly Liu et al.'s concept analysis[33]. The four broad areas (antecedents, attributes, consequences, and coping mechanisms) served as a theoretical framework derived from existing caregiver burden literature to guide our initial interview questions. However, the specific themes and subthemes emerged inductively from the data analysis. This approach represents framework-informed thematic analysis rather than pure inductive thematic analysis, allowing us to examine established concepts within a new cultural context while remaining open to novel findings. We analyzed data from semi-structured participant interviews, adhering to COREQ guidelines. This approach allows for an in-depth exploration of the lived experiences of family caregivers, capturing the essence of their caregiving journey [34,35]. Revised Results Four main themes emerged from the data, which aligned with but extended Liu et al.'s conceptual framework: antecedents to the caregiving burden, the attributes of the caregiving burden, the consequences of the burden, and religion and beliefs. Notably, our analysis revealed a novel fourth theme - 'religion and beliefs' - which was not identified in Liu et al.'s concept analysis, representing a unique contribution specific to our Saudi Arabian context. Additionally, we identified 'unsatisfied treatment' as a new subtheme under attributes, reflecting cultural expectations for family involvement in medical decisions. |
It is unusual for a qualitative study to have a similar outcome to an outcome from another article, especially when conducting a thematic analysis. This study has exactly similar wording and outcomes to another study (doi: 1016/j.ijnss.2020.07.012), which could lead to plagiarism and a questionable thematic analysis approach. Please provide a justification for this.
|
Citation standards All citations have been reviewed and formatted according to journal standards. Duplicated inclusion criteria We have eliminated the redundant mention of "caring for a relative with a cancer diagnosis of more than 2 weeks" in the inclusion criteria. we already upload Supplementary table for participant characteristics We have included the complete S1 Table with detailed participant characteristics in the supporting files section. We define abbreviations All abbreviations are now defined at first mention: Kingdom of Saudi Arabia (KSA), Family Caregivers (FCGs), etc. We have added to the Methods section: Data saturation was systematically assessed following established guidelines. After the 8th interview, no new codes emerged from the data. Interviews 9 and 10 confirmed theoretical saturation, with themes well-developed and no additional insights identified. This saturation point was documented through our analytical audit trail, which tracked code emergence across interviews. |
Minor comments Please pay attention to the citation standards and refer to them.
|

Reviewer 2 Report
Comments and Suggestions for Authors
Thank you for the opportunity to review your manuscript titled: “Developing burden caring for a relative with cancer diagnosis: A qualitative study of the lived experience of family caregivers in Saudi Arabia.” It’s an important topic, and to make improvements its vital to take note of caregiver perspectives as you have done, and especially how caregiving is experienced through various cultures. The discussion of religion is especially noteworthy in the findings. I have some concerns and comments mainly regarding the methodology which need to be clarified to help readers get a clear and transparent sense of what steps were taken to arrive at the findings. I note these and some points about accuracy in the writing below:
Abstract
These are some minor points with writing: There seems to be a missing word between “hidden patients” and family caregivers. Can you clarify? Also, it may read more clearly to say “Education and support for family caregivers may …” instead of using gerunds. (line 26)
Introduction
The second sentence cites the WHO about a prediction of cancer deaths that will occur in 2020. Since it’s now 2025, it would be more accurate to update this. What is a more recent prediction or what are some current estimates on cancer deaths in recent years?
Starting line 46, I’m wondering if it would be more accurate to convey experience as being imbued with meaning, rather than the other way around. Can you clarify the line of thought in that sentence?
The description of phenomenology starting around line 53 seems a little broad (thematic analysis does much of this too). Also, this type of method would not be used for evaluating experiences, especially with the emphasis to consider researcher bias. It would add accuracy to your work to provide a more distinct description of phenomenology, though see my note below. From the description of your analysis, this study seems to be more an example of thematic analysis.
The discussion of caregiving in Saudia Arabia is very informative and interesting. I was curious if the authors could clarify how social support is limited as noted on line 69 while families often share caregiving responsibilities (line 74-75). How do you see those as different? Or possibly do you mean that social support even with family help is limited for female caregivers?
The language for sentences from line 80-85 is a little unclear. Possibly there are some missing words, but can you clarify the meaning of social support patient and caregiver characteristics? Also, what is the connection related to caregiving responsibilities and then recognized factors like self-esteem and gender? Meaning, how have responsibilities been modified and what is meant more specifically by less recognized factors of self-esteem and gender?
Careful of typos and issues with grammar in certain areas, such as the last paragraph on page 2. The authors could also strengthen the rationale for conducting a qualitative study in this context by trimming down statements that seem to repeat the same idea.
Method
Under Setting – Can you give a more specific sense of what the authors mean when saying the hospital cares for many patients? Are there average numbers of patients per year or month you could report? Careful of the grammar in that section.
Inclusion criteria – I was curious about the rationale for selecting care partners who had provided care to a relative for more than 2 weeks. (line 152 and 160) It seems the experience of someone providing care for less than a month would be vastly different than someone doing so for closer to a year or longer. Can you discuss somewhere in the paper the rationale for chosen length of caregiving time (and possibly a large difference in range of time) and how it might play a role in caregiver perspectives? This could be a limitation though I read in that section a limitation being that the study focuses on a limited timeframe. By that, is it meant that the study looked at caregivers who all provided care for just around two weeks? Can you clarify?
Under data collection instruments, it’s stated that “the credibility, dependability, confirmability, and transferability of qualitative data are used to assess its validity and reliability.” This statement is a little vague and could use further explanation of what is being done and why.
More to the point, in this section, the interview is described as the data collection instrument, which I’m taking is meant as the interview guide being the instrument. It’s noted that several experts examined the instrument (interview questions/guide?) to check for the method’s validity. That wording is a little unclear since this is not referring to a quantitative instrument to be used to measure a specific construct. Can you describe more how the interview questions were developed and what specifically the experts were evaluating and for what purpose? I see the next section describes the questions as being created to examine 4 areas related to caregiving burden, so perhaps it’s more a matter of clarifying the above points in the language earlier on. Regardless, it would help readers understand the study’s process by describing the rationale for pursuing these four areas.
It’s also unclear in that the four areas discussed as interview questions are also later presented as themes that emerged from the data. The questions in the table read more like research questions, which suggests maybe they were predetermined themes the authors were interested in and used them to develop interview questions around. If so, that’s acceptable, though that needs to be described more clearly in the paper. Also, the method of using phenomenology to attempt to discover emerging themes would not truly apply. This discrepancy between questions and themes needs to be more clearly explained.
Around line 196 it’s stated that the pre-planned interview guide was created to ensure inter-rater consistency. Can you clarify what is being rated? For a semi-structured interview, if you are asking open-ended questions, how does inter-rater consistency play a role during an interview? Perhaps this is meant more for coding process which is mentioned later?
Data analysis
The first paragraph of this section describes more data collection rather than analysis.
Did all researchers participate in the coding? Can you elaborate there a bit more on how codes for analysis were originally developed (inductively vs deductively for example) and what were some of your codes? The manuscript implies transcripts were coded and themes developed one transcript at a time at first. (e.g., discussion of data saturation guiding the sample selection, line 153). But the section starting around line 216 suggests interviews were all coded and then themes developed. Can you clarify?
More importantly, the methods shift from proposing to conduct interpretive phenomenology to thematic analysis. These are two separate qualitative methods, although some tactics do overlap. It’s really important to clarify what your approach was and be consistent throughout the paper. Your overall approach seems to be more thematic analysis.
This relates back to an earlier comment, but on line 237 it’s mentioned there was thorough questioning by interviewers. Can you explain more since Table 1 only offers four broad categories. Were probing questions used? Can you give examples of some of the specific questions? Discussion of how the interviews were conducted relates more to data collection.
Can you provide more details regarding the process for using “ongoing comparison and analytical induction methods”? What is being compared and what specific induction methods are you using and how?
It was helpful that the authors included a statement of the researcher’s positionality, although the stated assumption that “gender can affect participant comfort levels” seemed to be a stereotype, especially with the claim not being cited. Can you clarify what other studies you were referring to that demonstrated this, if this is the case? Also, it’s not clear what is meant by “ the researcher generated authentic representations of the participants viewpoints”…For example how so? It could help to review other studies that have used this approach to see more examples of how researchers describe their positionality and its influence on data collection/analysis. For example, in some studies positionality is described in terms of researchers level of experience with a particular method and/or similarities or differences with study participants.
Here's a study that gives a pretty thorough discussion of researchers’ positionality:
Mayo AM, Siegle K, Savell E, Bullock B, Preston GJ, Peavy GM. Lay caregivers' experiences with caring for persons with dementia: A phenomenological study. Journal of gerontological nursing. 2020 Aug 1;46(8):17-27.
Results
I was curious from the participant discussions on self-perception, how this related to burden? Causal relationships can’t be assumed, yet can you clarify in this part how caregivers perceive self-perception as an aspect of burden?
Discussion
This is a minor point but by stating a goal to identify factors contributing to burden, “including”, financial strain, etc., it reads as if these were in mind prior to data collection. Using wording like “such as” can slightly shift the meaning to imply you are recalling aspects found in the data.
It could strengthen this study’s contribution to emphasize what the findings uncover that’s different from or that extends other work, beyond what research the findings align with. The caregiver perspectives related to religion contribute something new to existing research.
I don’t see the sample size as a limitation in that the goal of qualitative work is not to generalize findings but instead provide a richer understanding of unique experiences. Same thing with self-reported data. If the goal is to understand caregivers’ perspectives and experiences, and from their unique point of view, self-reported data is required. Questioning the validity of participant responses is not appropriate and could introduce researcher bias.
Some references seem to be missing information (e.g., # 33)
Comments on the Quality of English LanguageI pointed out some specifics in the earlier section, though overall just watch out for repeating ideas and some places where the phrasing gets a little vague.
Author Response
Title: Developing burden caring for a relative with cancer diagnosis: a qualitative study of the lived experience of family caregivers in Saudi Arabia
Thank you very much for the editors' and reviewers' comprehensive and constructive feedback on our manuscript. We appreciate the time and effort you have dedicated to reviewing our work. Below, we address each of your comments systematically:
|
|
RESPONSE |
Reviewer 2 |
Family caregivers are crucial in caring for and supporting cancer patients. Thus, it is essential to shed light on family caregivers, who often remain invisible as secondary patients in healthcare systems, to understand the factors associated with developing caregiver burden. Education and support for family caregivers may decrease the burden on the family caregiver, which can be reflected positively on the whole family unit, including the relative with a cancer diagnosis. |
Abstract These are some minor points with writing: There seems to be a missing word between “hidden patients” and family caregivers. Can you clarify? Also, it may read more clearly to say “Education and support for family caregivers may …” instead of using gerunds. (line 26)
|
Thank you for this important observation. We will update the cancer statistics with more recent data. The revised text will read: According to the World Health Organization (2024), by 2050, almost 35 million additional cancer cases are anticipated, representing a 77% rise over the projected 20 million cases in 2022[1].
Thank you We revised this sentence in line 46, for greater clarity: Revised text: Experience is a subjective phenomenon through which individuals imbue meaning into their perception and conscious response to a specific event or occurrence. Since the experience is subjective, applying any metrics to evaluate an individual's experience is impossible.
Thank you for this critical observation. You are correct that our methodology more accurately reflects thematic analysis rather than interpretative phenomenological analysis. Revised text: In Saudi Arabia, despite several studies on caregivers of cancer patients, research attempts to assess the impact of formal care on these individuals have not effectively yielded empirical insights into the true character or substance of the caregiving experience. Quantitative studies cannot comprehensively depict caring experiences as effectively as qualitative research studies. Thematic analysis, a qualitative methodology, enables researchers to thoroughly investigate, evaluate, and articulate the real-life encounters of those who care for cancer patients while identifying patterns and themes within their experiences."
Thank you for seeking this clarification. We revised this section to explain that, In Saudi Arabia, the burden on caregivers is exacerbated by limited formal social support systems, despite the significant impact on their quality of life. While families often share caregiving responsibilities among extended family members, social support remains constrained because: (1) the support is typically concentrated within the immediate family without broader community resources, (2) female caregivers may experience additional burden despite family help due to cultural expectations of primary caregiving responsibility, and (3) formal support systems remain inadequate regardless of informal family assistance. Caregivers often experience high levels of stress and anxiety due to these inadequate formal support systems
Comment 5: Language unclear in sentences from lines 80-85 regarding social support patient and caregiver characteristics. We clarified this section as follows: According to Williams et al. [25], characteristics of both patients and caregivers, along with available social support, influence the caregiving experience. Recent healthcare shifts have modified traditional caregiving responsibilities, while factors such as caregiver self-esteem and gender—which have received less research attention—also significantly affect burden levels.
We acknowledge these issues and conducted a thorough review to eliminate grammatical errors and redundant statements, particularly in the last paragraph of page 2. |
Introduction The second sentence cites the WHO about a prediction of cancer deaths that will occur in 2020. Since it’s now 2025, it would be more accurate to update this. What is a more recent prediction or what are some current estimates on cancer deaths in recent years?
Starting line 46, I’m wondering if it would be more accurate to convey experience as being imbued with meaning, rather than the other way around. Can you clarify the line of thought in that sentence?
The description of phenomenology starting around line 53 seems a little broad (thematic analysis does much of this too). Also, this type of method would not be used for evaluating experiences, especially with the emphasis to consider researcher bias. It would add accuracy to your work to provide a more distinct description of phenomenology, though see my note below. From the description of your analysis, this study seems to be more an example of thematic analysis.
The discussion of caregiving in Saudia Arabia is very informative and interesting. I was curious if the authors could clarify how social support is limited as noted on line 69 while families often share caregiving responsibilities (line 74-75). How do you see those as different? Or possibly do you mean that social support even with family help is limited for female caregivers?
The language for sentences from line 80-85 is a little unclear. Possibly there are some missing words, but can you clarify the meaning of social support patient and caregiver characteristics? Also, what is the connection related to caregiving responsibilities and then recognized factors like self-esteem and gender? Meaning, how have responsibilities been modified and what is meant more specifically by less recognized factors of self-esteem and gender?
Careful of typos and issues with grammar in certain areas, such as the last paragraph on page 2. The authors could also strengthen the rationale for conducting a qualitative study in this context by trimming down statements that seem to repeat the same idea.
|
The study was conducted in the Prince Sultan Military Medical City's oncology department in Riyadh, a governmental hospital in the Kingdom of Saudi Arabia affiliated with the Ministry of Health and the Ministry of Defense and Affairs (MODA) in Riyadh City, KSA. It is a regional hospital with 192 beds that serves approximately 14,300outpatients annually in all departments, with an average of 100 patients per month receiving cancer treatment and follow-up care.
Thank you for this important point. We added a limitation acknowledging that the wide range of caregiving duration (2 weeks to several months) may have influenced participants' perspectives. We chose the 2-week minimum to ensure participants had sufficient exposure to caregiving demands while recognizing that this criterion may not capture the full spectrum of long-term caregiving experiences. This will be explicitly discussed as a limitation.
Thank you. We provided more specific details: To establish trustworthiness, we employed multiple strategies: (1) the interview guide was reviewed by five experts (mental health nursing faculty, psychiatrist, psychologist, social worker, and mental health nurse) who evaluated content validity and cultural appropriateness; (2) questions were piloted with three caregivers to assess clarity and relevance; (3) member checking was conducted with participants to validate emerging themes; and (4) an audit trail documented all analytical decisions.
Thank you. We clarified that the four broad areas (antecedents, attributes, consequences, and coping mechanisms) served as a theoretical framework derived from existing caregiver burden literature to guide our initial interview questions. However, the specific themes and subthemes emerged inductively from the data analysis. This approach represents a form of framework analysis rather than pure inductive thematic analysis, which we will explicitly state and justify.
Thank you. We clarified the text Inter-rater consistency refers to the coding process, where multiple researchers independently coded transcripts and compared their interpretations until consensus was reached, rather than consistency during interviews. |
Method Under Setting – Can you give a more specific sense of what the authors mean when saying the hospital cares for many patients? Are there average numbers of patients per year or month you could report? Careful of the grammar in that section.
Inclusion criteria – I was curious about the rationale for selecting care partners who had provided care to a relative for more than 2 weeks. (line 152 and 160) It seems the experience of someone providing care for less than a month would be vastly different than someone doing so for closer to a year or longer. Can you discuss somewhere in the paper the rationale for chosen length of caregiving time (and possibly a large difference in range of time) and how it might play a role in caregiver perspectives? This could be a limitation though I read in that section a limitation being that the study focuses on a limited timeframe. By that, is it meant that the study looked at caregivers who all provided care for just around two weeks? Can you clarify?
Under data collection instruments, it’s stated that “the credibility, dependability, confirmability, and transferability of qualitative data are used to assess its validity and reliability.” This statement is a little vague and could use further explanation of what is being done and why.
More to the point, in this section, the interview is described as the data collection instrument, which I’m taking is meant as the interview guide being the instrument. It’s noted that several experts examined the instrument (interview questions/guide?) to check for the method’s validity. That wording is a little unclear since this is not referring to a quantitative instrument to be used to measure a specific construct. Can you describe more how the interview questions were developed and what specifically the experts were evaluating and for what purpose? I see the next section describes the questions as being created to examine 4 areas related to caregiving burden, so perhaps it’s more a matter of clarifying the above points in the language earlier on. Regardless, it would help readers understand the study’s process by describing the rationale for pursuing these four areas. It’s also unclear in that the four areas discussed as interview questions are also later presented as themes that emerged from the data. The questions in the table read more like research questions, which suggests maybe they were predetermined themes the authors were interested in and used them to develop interview questions around. If so, that’s acceptable, though that needs to be described more clearly in the paper. Also, the method of using phenomenology to attempt to discover emerging themes would not truly apply. This discrepancy between questions and themes needs to be more clearly explained. Around line 196 it’s stated that the pre-planned interview guide was created to ensure inter-rater consistency. Can you clarify what is being rated? For a semi-structured interview, if you are asking open-ended questions, how does inter-rater consistency play a role during an interview? Perhaps this is meant more for coding process which is mentioned later?
|
We relocated the data collection information to the appropriate section and focus this paragraph solely on analytical procedures.
We clarified coding process and researcher participation: Both authors participated in the coding process. Initial codes were developed inductively from the data, with researchers independently coding the first three transcripts and then meeting to compare codes and develop a preliminary coding framework. Examples of initial codes included 'financial strain,' 'role conflict,' 'physical exhaustion,' and 'spiritual coping.
As mentioned earlier, we revised throughout the manuscript to accurately reflect that we conducted thematic analysis, acknowledging that our focus on lived experiences aligns with phenomenological interests but our analytical approach follows Braun & Clarke's thematic analysis framework.
We added: While Table 1 presents the four main question areas, interviewers used extensive probing questions such as 'Can you tell me more about that?' 'How did that make you feel?' and 'What was that experience like for you?' to explore participants' responses in depth.
We clarified the text: Ongoing comparison involved continuously comparing new data with previously coded data to refine themes and identify patterns. Analytical induction involved systematically examining cases that did not fit emerging patterns to ensure comprehensive theme development.
We clarified this section to provide more specific information about researcher positionality without unsupported assumptions: The lead researcher's position as a female psychiatric nurse from a similar cultural background facilitated participant comfort and cultural understanding. However, we remained aware that this similarity could potentially influence data interpretation and employed member checking and peer debriefing to ensure authentic representation of participant perspectives. |
Data analysis The first paragraph of this section describes more data collection rather than analysis.
Did all researchers participate in the coding? Can you elaborate there a bit more on how codes for analysis were originally developed (inductively vs deductively for example) and what were some of your codes? The manuscript implies transcripts were coded and themes developed one transcript at a time at first. (e.g., discussion of data saturation guiding the sample selection, line 153). But the section starting around line 216 suggests interviews were all coded and then themes developed. Can you clarify?
More importantly, the methods shift from proposing to conduct interpretive phenomenology to thematic analysis. These are two separate qualitative methods, although some tactics do overlap. It’s really important to clarify what your approach was and be consistent throughout the paper. Your overall approach seems to be more thematic analysis.
This relates back to an earlier comment, but on line 237 it’s mentioned there was thorough questioning by interviewers. Can you explain more since Table 1 only offers four broad categories. Were probing questions used? Can you give examples of some of the specific questions? Discussion of how the interviews were conducted relates more to data collection.
Can you provide more details regarding the process for using “ongoing comparison and analytical induction methods”? What is being compared and what specific induction methods are you using and how?
It was helpful that the authors included a statement of the researcher’s positionality, although the stated assumption that “gender can affect participant comfort levels” seemed to be a stereotype, especially with the claim not being cited. Can you clarify what other studies you were referring to that demonstrated this, if this is the case? Also, it’s not clear what is meant by “ the researcher generated authentic representations of the participants viewpoints”…For example how so? It could help to review other studies that have used this approach to see more examples of how researchers describe their positionality and its influence on data collection/analysis. For example, in some studies positionality is described in terms of researchers level of experience with a particular method and/or similarities or differences with study participants. Here's a study that gives a pretty thorough discussion of researchers’ positionality: Mayo AM, Siegle K, Savell E, Bullock B, Preston GJ, Peavy GM. Lay caregivers' experiences with caring for persons with dementia: A phenomenological study. Journal of gerontological nursing. 2020 Aug 1;46(8):17-27. |
Thank you. We revised the text : Self-perception emerged as an attribute of burden in that caregivers' feelings about their caregiving role and competence directly influenced their experience of burden. Positive self-perception (feeling capable and satisfied with caregiving) was associated with lower perceived burden, while negative self-perception (feeling inadequate or overwhelmed) corresponded with higher burden levels.
|
Results I was curious from the participant discussions on self-perception, how this related to burden? Causal relationships can’t be assumed, yet can you clarify in this part how caregivers perceive self-perception as an aspect of burden?
|
We made this change to better reflect that these factors emerged from our data rather than being predetermined.
We expand the discussion of how our findings extend existing knowledge, particularly emphasizing the unique contribution of the religious coping theme in the Saudi Arabian context.
Thank you. We agree with your perspective and will remove these from the limitations section, focusing instead on more substantive limitations such as the specific cultural context, convenience sampling, and the range of caregiving duration.
We corrected reference #33 to include complete citation information. |
Discussion This is a minor point but by stating a goal to identify factors contributing to burden, “including”, financial strain, etc., it reads as if these were in mind prior to data collection. Using wording like “such as” can slightly shift the meaning to imply you are recalling aspects found in the data.
It could strengthen this study’s contribution to emphasize what the findings uncover that’s different from or that extends other work, beyond what research the findings align with. The caregiver perspectives related to religion contribute something new to existing research.
I don’t see the sample size as a limitation in that the goal of qualitative work is not to generalize findings but instead provide a richer understanding of unique experiences. Same thing with self-reported data. If the goal is to understand caregivers’ perspectives and experiences, and from their unique point of view, self-reported data is required. Questioning the validity of participant responses is not appropriate and could introduce researcher bias. Some references seem to be missing information (e.g., # 33)
|
We conducted a comprehensive review to address grammatical issues and improve clarity throughout the manuscript. |
Comments on the Quality of English Language
|
- 1- World Health Organization (WHO). Global cancer burden growing, amidst mounting need for services [Internet]. News release. 2024 [cited 2025 May 28]. Available from: https://www.who.int/news/item/01-02-2024-global-cancer-burden-growing--amidst-mounting-need-for-services
Kind regards,

Round 2
Reviewer 1 Report
Comments and Suggestions for Authors
The authors provided an explanation of the reviewer's concern about the potential plagiarism from the previous review. Improvements have also been made to improve the quality of the paper. Some remaining comments:
- Some suggestions, although they had been addressed in the author's reply, are still not shown in the paper. e.g. conclusion sentences, abbreviations.
- Ensure consistency throughout the paper. e.g. the use of the word family caregiver/FCG/FCGs.
- To improve the flow of the introduction, I suggest moving the sentences from lines 54-63 to under line 112.
Author Response
Title: Developing burden caring for a relative with cancer diagnosis: a qualitative study of the lived experience of family caregivers in Saudi Arabia
Thank you very much for the reviewers' comprehensive and constructive feedback on our manuscript. We appreciate the time and effort you have dedicated to reviewing our work. Below, we address each of your comments systematically:
Reviewers comments |
|
RESPONSE |
reviewer 1
|
Thank you for pointing this out. we have added the missing elements: “This study contributes to the growing body of literature on caregiver burden by identifying culturally specific factors within the Saudi context and proposing a framework for culturally sensitive interventions” and also revised the conclusion Thank you we have reviewed entire manuscript to ensure consistent usage FCGs.
Thank you for your respectful suggestion, we have restructured the introduction as suggested |
The authors provided an explanation of the reviewer's concern about the potential plagiarism from the previous review. Improvements have also been made to improve the quality of the paper. Some remaining comments: · Some suggestions, although they had been addressed in the author's reply, are still not shown in the paper. e.g. conclusion sentences, abbreviations. · Ensure consistency throughout the paper. e.g. the use of the word family caregiver/FCG/FCGs. To improve the flow of the introduction, I suggest moving the sentences from lines 54-63 to underline 112 |
Regards,

Reviewer 2 Report
Comments and Suggestions for Authors
Thank you for the revisions made to the manuscript as they address many of the points of interest I had. I have just three things to bring to your attention from this revised paper:
- In shifting the type of analysis to thematic from phenomenological, be sure to make that adjustment in the abstract. The abstract still mentions phenomenological methods (along with thematic analysis).
- Line 62, qualitative studies can't evaluate people's experiences. They can identify people's evaluation of a particular phenomenon, but to suggest the former can imply unwarranted bias. I'd adjust the phrasing in that section to avoid indicating a goal of evaluation.
- Similarly, in the results, around line 408, it's misleading to suggest caregiver self-perception was associated with burden as no quantitative tests were run to test this. It's not clear from the shared excerpts what the relationship is between how they perceive themselves and the burden they experience. This connection needs to be more clearly explained by finding participant statements where they more explicitly discuss self-perception and burden together.
Author Response
Title: Developing burden caring for a relative with cancer diagnosis: a qualitative study of the lived experience of family caregivers in Saudi Arabia
Thank you very much for the reviewers' comprehensive and constructive feedback on our manuscript. We appreciate the time and effort you have dedicated to reviewing our work. Below, we address each of your comments systematically:
Reviewers comments |
|
Response |
Reviewer 2
|
Thank you for your comments, we have corrected the abstract to reflect the thematic analysis approach
we have revised the problematic phrasing: Thematic analysis, a qualitative methodology, enables researchers to thoroughly investigate, explore, and articulate the real-life encounters of those who care for cancer patients while identifying patterns and themes within their experiences [29].
Thank you for valuable comments, we have expanded this section with more explicit participant statements and clearer explanation The participants expressed their self-perceptions or feelings about themselves caring for a relative with a cancer diagnosis. Self-perception is how caregivers perceive themselves while caring for a relative with a cancer diagnosis or what feelings they have about themselves and their new role as caregivers. Positive self-perception (feeling capable and satisfied with caregiving) was associated with lower perceived burden, while negative self-perception (feeling inadequate or overwhelmed) corresponded with higher burden levels, as evidenced by contrasting participant statements. One participant expressing positive self-perception stated, 'My feeling about my experience is alhamdulillah good' (P5), while another participant expressing negative self-perception said, 'We are only watching my mother dying' (P4). These contrasting perspectives emerged from participants' narratives about their caregiving experiences, with those expressing competence and spiritual acceptance reporting less distress compared to those expressing helplessness and despair.
|
Thank you for the revisions made to the manuscript as they address many of the points of interest I had. I have just three things to bring to your attention from this revised paper:
|
Regards,
